# The Impact of Infused Autograft Absolute Numbers of Immune Effector Cells on Survival Post-Autologous Stem Cell Transplantation

**DOI:** 10.3390/cells11142197

**Published:** 2022-07-14

**Authors:** Luis F. Porrata

**Affiliations:** Division of Hematology, Department of Internal Medicine, Mayo Clinic, Rochester, MN 55905, USA; porrata.luis@mayo.edu; Tel.: +1-507-284-5096; Fax: +1-507-266-4972

**Keywords:** autograft immune effector cells, autologous stem cell transplantation, survival

## Abstract

Autologous stem cell transplantation treatment has been viewed as a therapeutic modality to enable the infusion of higher doses of chemotherapy to eradicate tumor cells. Nevertheless, recent reports have shown that, in addition to stem cells, infusion of autograft immune effector cells produces an autologous graft-versus-tumor effect, similar to the graft-versus-tumor effect observed in allogeneic-stem cell transplantation, but without the clinical complications of graft-versus-host disease. In this review, I assess the impact on clinical outcomes following infusions of autograft-antigen presenting cells, autograft innate and adaptive immune effector cells, and autograft immunosuppressive cells during autologous stem cell transplantation. This article is intended to provide a platform to change the current paradigmatic view of autologous stem cell transplantation, from a high-dose chemotherapy-based treatment to an adoptive immunotherapeutic intervention.

## 1. Introduction

The current dogma regarding the mechanism of action of infused donor allogeneic-reactive immune effector cells is that they produce a graft-versus-tumor effect, thereby targeting tumor cells in different types of allogeneic stem cell transplantation: full-myeloablative, reduced-intensity, cord, or haploidentical. Evidence for the clinical benefit of this graft-versus-tumor effect has been derived from indirect observations, including the following: (1) the withdrawal of immunosuppression following disease relapse post-allogeneic stem cell transplantation re-establishes remission; (2) a high relapse risk has been observed in syngeneic transplants compared with allogeneic transplants; (3) the development of graft-versus-host disease (GVHD) has been associated with lower relapse rates; (4) T cell infusion depletes the allograft, resulting in a higher risk of relapse; and (5) relapses post-allogeneic stem cell transplantation respond to donor lymphocyte infusion [1]. A fundamental problem with infused allogeneic-reactive immune effector cells is their lack of specificity, as the immunocompetent donor cells not only target the tumor through the graft-versus-tumor effect, but also the host, producing GVHD. GVHD is one of the main contributors to the high morbidity and transplant-related mortality seen in the allogeneic stem cell transplantation setting [1]. Therefore, continuous efforts are ongoing to decrease GVHD without affecting the clinical benefits of the graft-versus-tumor effect.

In contrast, autologous stem cell transplantation (ASCT) is mainly regarded as a way to combat the effects of high-dose chemotherapy, with the chemotherapy itself considered the main mechanism of action, eliminating cancer cells that have evaded standard doses of chemotherapy. ASCT has not been favorably viewed as an immunotherapy modality compared with allogeneic stem cell transplantation or more recently developed chimeric antigen receptor T-cell therapies. However, a large body of publications in the medical literature is challenging this dogma to also view ASCT as an immunotherapeutic modality and not just a means to administer higher doses of chemotherapy. The development of an autologous graft-versus-tumor effect using patients’ own immune system would bypass the detrimental effects of GVHD seen with allogeneic stem cell transplantation. In this article, I assess the current clinical data supporting the concept of an autologous graft-versus-tumor effect by reviewing the different immune effector cells that are collected and infused in the autograft and their association with survival post-ASCT.

## 2. Absolute Lymphocyte Count Recovery Post-Autologous Stem Cell Transplantation

Our group was the first to report that absolute lymphocyte count recovery at day 15 (ALC-15) ≥ 500 cells/µL after autologous stem cell transplantation in patients with B-cell non-Hodgkin’s lymphoma and multiple myeloma was associated with superior overall survival (OS) and progression-free survival (PFS) [2]. The superior OS and PFS based on ALC-15 were subsequently confirmed in retrospective studies in patients treated with autologous stem cell transplantation for B-cell non-Hodgkin’s lymphoma [3,4], T-cell non-Hodgkin’s lymphoma [5], mantle cell lymphoma [6], Hodgkin’s lymphoma [3,7], acute myelogenous leukemia [8], multiple myeloma [9], systemic amyloidosis [10], breast cancer [11,12], and ovarian cancer [13]. Although retrospective analysis is subject to its accompanying biases, two prospective studies in B-cell non-Hodgkin’s lymphoma have also detected better survival with an ALC-15 ≥ 500 cells/µL [14,15]. These reports of superior survival post-ASCT based on the ALC-15 in multiple different malignancies provided the first clinical evidence that patients’ own immunity has anti-tumor activity without the detrimental side effect of GVHD in allogeneic stem cell transplantation or the cytokine release syndrome reported in chimeric antigen receptor T-cell therapy [16,17].

## 3. Sources of Absolute Lymphocyte Recovery at Day 15 Post-Autologous Stem Cell Transplantation

The discovery of ALC-15 as a prognostic factor in survival post-ASCT suggests that the identification of sources affecting the ALC-15 could enable their use as an immunotherapeutic strategy to improve clinical outcomes in the ASCT setting. Possible sources for the ALC-15 in the ASCT setting include: (1) host stem cells and lymphocytes surviving the high-dose chemotherapy; and (2) the infusion of stem cell (CD34) or the infusion of the autograft-absolute lymphocyte count (A-ALC) [1]. The host stem cells surviving the high-dose chemotherapy are not the main source of ALC-15 because, without stem cell rescue, patients remained myelosuppressed for prolong periods of time. Compared with allogeneic stem cell transplantation, where the use of chimerism analysis can determine cells for donor and the host, in the ASCT we do not approve clinical techniques to tag and follow the recovery of autologous lymphocytes that survive the high-dose chemotherapy.

Our group evaluated whether the dose of infused CD34 stem cells correlated with ALC-15 and no correlation was identified in lymphoma patients (Spearman’s rho, *r* = 0.11, *p* = 0.8) or in multiple myeloma patients (Spearman’s rho, *r* = 0.27, *p* = 0.67) [18,19]. Other groups have also reported no association between the dose of infused CD34 stem cells and absolute lymphocyte count recovery post-ASCT [20]. Another source of the ALC-15 post-ASCT is the number of autografts collected and the infused lymphocytes. Currently, for ASCT, no manipulation is performed on the cellular components that are collected and infused to the patients, such as T-cell autograft depletion via CD34 stem cell selection. Therefore, all cellular components collected in the autograft are re-infused to the patients. Our group investigated if there was a direct correlation between A-ALC and ALC-15. We reported a positive correlation between A-ALC and ALC-15 in patients treated with ASCT for non-Hodgkin’s lymphoma (Spearman’s rho, *r* = 0.71, *p* < 0.0001) and multiple myeloma (Spearman’s rho, *r* = 0.83, *p* < 0.0001) [18,19]. This correlation between A-ALC and absolute lymphocyte count recovery post-ASCT has been confirmed by other groups in multiple myeloma [20].

## 4. Infusion of Autograft-Absolute Lymphocyte Count

Due to the association between A-ALC and ALC-15, we evaluated whether the A-ALC was also a prognostic factor for survival in patients undergoing ASCT. The A-ALC was also identified as a prognostic factor for OS and PFS [18,19]. Table 1 shows the survival benefits for patient infused with an A-ALC ≥ 0.5 × 10^9^ cells/kg compared with those who did not receive this treatment. The discovery of the A-ALC as a prognostic factor for survival in ASCT led to a phase III, double-blind, randomized clinical trial to assess if higher infusion numbers of A-ALC translated into better clinical outcomes. In the phase III clinical trial, the 2-years OS rates for patients infused with an A-ALC ≥ 0.5 × 10^9^ cells/kg was 90% versus 70% for patients infused with an A-ALC < 0.5 × 10^9^ cells/kg; and the 2-years PFS rates were 80% versus 50%, respectively [21]. A follow-up at median 10.6 years was conducted with all of the patients who were recruited and completed the phase III trial: the 13-years OS rate was 54% for the A-ALC ≥ 0.5 × 10^9^ cells/kg group and 28% for the A-ALC < 0.5 × 10^9^ cells/kg group (*p* < 0.0007). The 13-years PFS rate following A-ALC ≥ 0.5 × 10^9^ cells/kg was 46% versus 17% for those with an A-ALC < 0.5 × 10^9^ cells/kg [22]. However, patients still relapsed despite being infused with an A-ALC ≥ 0.5 × 10^9^ cells/kg.

## 5. Infusion of Autograft-Professional Antigen-Presenting Cells

A limitation of the A-ALC lies in that fact that it does not provide more information regarding what specific type of immune effector cells is associated with the survival benefit or disadvantage seen post-ASCT. Professional antigen-presenting cells are among the first immune effector cells triggering an immune response against foreign bodies (i.e., tumor cells). Among the professional antigen-presenting cells are dendritic cells. Dendritic cells are divided into dendritic cells type 1 (DC1) and type 2 (DC2) [23]. DC1 express CD11 and have a myeloid morphology. When DC1 are stimulated by tumor necrosis factor, DC1 produce high levels of interleukin 12 (IL-12) causing antigen-naïve CD4 to differentiate into T helper type 1 cells [23]. T helper type 1 cells secret interferon-gamma and tumor necrosis factor-alpha leading to the activation and development of cytotoxic T cells. Another mechanism by which DC1 develops cytotoxic T cells is by presenting exogenous, cell-associated antigens into major histocompatibility complex class 1, leading to the priming of CD8 T cells against tumors [24]. In addition, via the production of interferon type 1 (IFN-1), DC1 cells enhance the function and priming against cancer cells by natural killer (NK) cells [25]. Furthermore, activated DC1 can directly kill tumor cells through tumor necrosis factor-related apoptosis-including ligand (TRAIL) and Granzyme B-dependent mechanisms causing tumor regression [26]. In contrast, DC2 has been associated with the differentiation of antigen-naïve CD4 into T helper type 2 cells, leading to a more pro-inflammatory microenvironment and immunosuppression [23]. Dean et al. initially reported better survival in patients with higher numbers of collected and infused autograft-dendritic cells (A-DC) [27]. Dean et al. then reported that the autograft-dendritic cell type 1 was the main sub-type of the A-DC conferring the survival advantages [27]. Our group subsequently confirmed that the infusion of autograft dendritic cells type 1 affected clinical outcomes post-ASCT (see Table 2) [28].

## 6. Infusion of Autograft-Innate Immune System

As dendritic cells also interact with NK cells, our group investigated whether innate immune effector cells are also collected and infused from the autograft in patients undergoing ASCT. Recent interest has focused on studying how macrophages can target tumor cells. Macrophages are divided into macrophage type 1 and macrophage type 2 [29]. Macrophage type 1 is associated with tumor elimination via mechanisms dependent on reactive oxygen species and reactive nitrogen species, as well as production of Interleukin-1β and tumor necrosis factor-alpha. Macrophage type 1 activates T cells and NK cells, promoting indirect cytotoxicity against tumor cells. In contrast, Macrophage type 2 is associated more with immunosuppression, as macrophage type 2 is recruited by the tumor as a tumor-associated macrophage to help the tumor cells evade the immune system [29]. Clinical trials are ongoing using TTI-621 to block the CD47 anti-phagocytic (“do not eat”) signal used by tumor cells by binding to the SIRPα on macrophages to evade the phagocytosis of tumors [30]. Thus, we are the first group to report that both autograft macrophages type 1 and type 2 are collected and infused to patients undergoing ASCT [28]. Low numbers of infused autograft-macrophage type 2 were associated with better clinical outcomes post-ASCT (see Table 3) [28].

Another important member of the innate immune system is the NK cells. NK cells can target tumor cells without prior sensitization. Looking at absolute numbers of NK cells, patients infused with higher numbers of autograft-natural killer cells survived longer than those who received less (see Table 3) [31,32]. The recognition and elimination of tumor cells by NK cells is highly regulated by the different types of receptors expressed on the NK cell surface. The first group of receptors (killer cell immunoglobulin-like receptors [KIRs]) is activated when binding with major histocompatibility complex (MCH) class I molecules histocompatibility complex (HLA): HLA-A, -B, and -C [33]. The second receptor group belongs to the C-type lectin-like heterodimeric receptor NKG2A, which recognizes the non-classical I molecule HLA-E [33]. Natural cytotoxicity receptors are a type of activating receptor [33,34]. These natural cytotoxicity receptors are NKp46, NKp30, and NKp44. NKp46 and NKp30 are constitutively expressed [33,34]. Ligands for this natural cytotoxicity receptor have been detected in tumor cells. In allogeneic stem cell transplantation, the mismatch of NK cells to KIR receptors affects survival [35]. In the autologous setting, our group reported also that the infusion of autograft natural killer cells KIR (KIR2DL2-inhibitoring receptor) and autograft natural cytotoxicity receptor (NKp30) also affect survival post-ASCT (see Table 3) [28].

## 7. Infusion of Autograft-Adaptive Immune System

Looking at the total normal of T cells using the CD3 marker, higher number of infused autograft CD3 correlated with a better clinical outcome post-ASCT [36]. Furthermore, higher numbers of infused autograft T helper cells (CD4) and autograft T cytotoxic cells (CD8) were also associated with better survival [31,37,38]. The program cell death ligand 1 (PD-1L) is a marker of T-cell immunologic exhaustion leading to T-cell dysfunction in cancer patients. It is important to differentiate between T-cell exhaustion dysfunction and “anergy” [39]. In anergy, the absence of signal 2 leads to a failure of T-cell activation. In contrast, immunologic exhaustion occurs after T cells have been fully activated and are unable to perform their effector response (i.e., the elimination of tumor cells) [39]. PD-1L is highly expressed in multiple different tumor malignancies and the expression of PD-1L correlates inversely with prognosis and survival [39]. PD-1L-associated T-cell apoptosis is one of the many potential mechanisms, which is supported by the inverse association between PD-1L expression in the cancer cells and the number of tumor infiltrating T cells [39]. To our knowledge, our group is the first to report that autograft infusion of CD4-PD-1- (lack of immunologic exhaustion) resulted in better survival post-ASCT (see Table 4) [28].

## 8. Infusion of Autograft-Immunosuppressive Effector Cells

Another immune effector cell collected and infused in conjunction with stem cell is the autograft monocytes. It has been reported that monocytes may induced immune tolerance in the ASCT setting by secreting kynurenine causing T-cell inhibition through apoptosis [40]. Thus, we looked at the collection and infusion of autograft-absolute monocyte count to assess if the autograft-absolute monocyte count also impact survival post-ASCT. Table 5 depicts that patient infused with a higher number of autograft-absolute monocyte count experienced worse survival in comparison with patients infused with less numbers of autograft-absolute monocyte count [41]. To simplify the use of these two prognostic factors (autograft-absolute lymphocyte count and autograft-absolute monocyte count) to predict clinical outcomes post-ASCT, we combined the autograft-absolute lymphocyte (as surrogate marker of host immunity) and autograft-absolute monocyte count as surrogate marker of immunosuppression) into the autograft lymphocyte to monocyte ratio. Table 5 shows that patients infused with an autograft lymphocyte to monocyte ratio ≥ 1 had better OS and PFS for both B-cell non-Hodgkin’s lymphoma and T-cell non-Hodgkin’s lymphoma patients treated with ASCT.

Myeloid-derived suppressor cells (MDSC) are a heterogenous population of progenitor and myeloid cells involved in tumor associated immune suppression. Several immunosuppressive mechanisms attributed to MDSC include: (1) production of immunosuppressive cytokines (i.e., interleukin-10); (2) disruption of major histocompatibility complex class 1 receptor causing T cells to become unresponsive to antigen-specific interaction; (3) MDSC express the death toll receptor Fas-thus Fas-FasL interaction leading to T cell apoptosis; (4) induction of regulatory T cells; and (5) inhibition of NK cells function and proliferation [40,42]. A subset of MDSC, characterized as monocytic CD14^+^ cells with low levels of lack of the antigen presenting HLA-DR molecules (CD14^+^HLA-DR^low/neg (DIM)^) [43] has been associated as a negative prognostic factor for survival circulating in the peripheral blood of lymphoma patients [44]. Thus, our group also identified the collection and infusion of high numbers of autograft MDSC and CD14^+^HLA-DR^low/neg (DIM)^ cells correlated with worst clinical outcomes post-ASCT (see Table 5) [32].

## 9. Autologous Graft-Versus-Tumor Effect

The idea that ASCT can target cancer cells through an autologous graft-versus-tumor effect is considered a foreign concept. However, the cumulative medical literature argues otherwise. This current review identified autograft dendritic cells type 1, CD3, CD4, CD4-PD-1-, CD8, NK cells, and NKp30 NK cells as good prognostic factors for survival post-autologous stem cell transplantation. However, the immunosuppressive immune effector cells collected and infused in the autograft are associated with a worse clinical outcome post-ASCT: monocytic CD14+HLA-DR^DIM^ cells, KIR2DL2 NK cells, MDSC, and macrophages type 2. The significant finding that all of the infused autograft immune effectors affect survival (antigen-presenting cells = dendritic cells type 1; innate immune system = NK cells/NKp30; and adaptive immune cells = CD3, CD4, CD4-PD-1-, CD8); as well as the immuno-suppressive components (monocytic CD14+HLA-DR^DIM^ cells, KIR2DL2 NK cells, MDSC, and macrophages type 2) provides a broader platform to target the entire immune system in ASCT. Despite the recent advances of chimeric antigen receptor T-cell therapy, a limitation of this treatment modality is that it only targets one component of the immune system, i.e., the T cells of the adaptive immune system: hence, the problem of antigen escape. Antigen escape occurs when the tumor develops resistance to a single antigen-targeting chimeric antigen receptor T-cell therapy by either partial or complete loss of target antigen expression [16]. Examples include the downregulation/loss of CD19 antigen in B-cell acute lymphoblastic leukemia as well as the downregulation or loss of B-cell maturation antigen expression in multiple myeloma leading to resistance against chimeric antigen receptor T-cell therapy [16]. To bypass single antigen escape, dual-targeted chimeric antigen receptor T-cell therapy constructs (CD19/CD22 in B-cell acute lymphoblastic leukemia and B-cell maturation antigen/CD19 in multiple myeloma) are undergoing clinical trials [16]. Other limitations of chimeric antigen receptor T-cell therapy include: (1) on-target off-tumor effects (the targeted antigen in the tumor is also expressed in normal cells); and (2) an immunosuppressive microenvironment that can render the chimeric antigen receptor T cells ineffective (i.e., MDSC and tumor-associated macrophages) [16]. There is a similar problem with ASCT, as the infusion of autograft myeloid-derived cells along with autograft CD14+HLA-DR^DIM^ cells was associated with worse clinical outcomes.

In allogeneic stem cell transplantation, the complication of graft-versus-host disease continues to be a difficult issue, causing serious morbidities and mortality. Despite multiple therapeutic treatments to control graft-versus-host disease including steroids, Janus kinase 1 and 2 inhibitor (Ruxolitinib), inhibitor of purine synthesis (Mycophenolate mofetil), recombinant human tumor necrosis factor (TNF)-alpha receptor fusion protein (Etanercept), purine analog (Pentostatin), protease inhibitor (Alpha-1 antitrypsin), mammalian targets of rapamycin (Sirolimus), anti-interleukin-6 (Tocilizumab), extracorporeal photopheresis, and mesenchymal stromal cells, the mortality of patients developing grade 4 GVHD is high with an associated survival rate of 5% [45].

## 10. Autologous Graft-Versus-Tumor Effect Engineering

Schematic representation presents a summary of the current understanding of the impact of autograft immune effector cell collection and infusion in improving or decreasing survival post-ASCT. Based on this knowledge, new methods are warranted to enhance the collection of pro-survival autograft immune effector cells (i.e., dendritic cells type 1, CD4, CD4-PD-1-, CD8, NK cells, and NKp30 cells) and to eliminate the collection and infusion of immuno-suppressive autograft immune effector cells (i.e., macrophage type 2, myeloid-derived suppressor cells, and CD14+HLA-DR^DIM^).

Two strategies to engineer an autologous graft-versus-tumor effect are the mobilization and the collection of the desired immune effector cells.

### 10.1. Mobilization of Immune Effector Cells

Interleukins are a type of cytokine that function in the activation, differentiation, proliferation, maturation, migration, and adhesion of immune cells. Among these cytokines, candidates for immune effector cell mobilization include interleukin-2 (IL-2), interleukin-15 (IL-15), and interleukin-21 (IL-21).

Interleukin-2: In a pilot study of patients with breast cancer, IL-2 was combined with granulocyte-colony-stimulating-factor (G-CSF) to mobilize immune effector cells. Patients received IL-2 two weeks before stem cell mobilization and collection. Higher NK cell recovery 14 days post-ASCT was observed in the mobilized group with IL-2 and G-CSF compared with the mobilized group with G-CSF alone [46]. However, IL-2 has been associated with the proliferation of regulatory T cells. Regulatory T cells have been associated with the downregulation of the immune response. This is significant because lymphoma patients who showed high proliferation of regulatory T cells by day 28+ post-ASCT experienced higher mortality and worse OS [47].

Interleukin-15: IL-15 is an important cytokine for NK cell proliferation. Our group reported that high IL-15 expression by day 15 post-ASCT was associated with higher NK cell recovery and better survival [48]. In the allogeneic stem cell transplantation setting [48], the use of the IL-15 super-agonist ALT-803 has been shown to stimulate the activation, proliferation, and expansion of NK cells and CD8+ T cells, with no increase in regulatory T cells [49].

Interleukin-21: IL-21 is another cytokine that affects the maturation and proliferation of natural killer cells. IL-21 could be combined with G-CSF to affect the mobilization and collection of natural killer cells [50].

Granulocyte-colony-stimulating-factor: Since the development of peripheral blood stem cell mobilization and collection, G-CSF has been and continues to be the main drug used to mobilize stem cells. However, G-CSF has been associated with the generation of regulatory T cells and MDSC [43,51]. Therefore, the use of G-CSF could explain why we reported MDSC/CD14+HLA-DR^DIM^ collected in the autograft. The development of drugs that minimize the mobilization and collection of immuno-suppressive cells is warranted.

Chemotherapy mobilization: If G-CSF alone fails to mobilize enough stem cells, G-CSF has been combined with chemotherapy to enhance stem cell mobilization and collection. However, it has been reported that, due to the immunosuppressive effects of chemotherapy, lower numbers of lymphocytes are collected in the autograft compared with patients who are mobilized with G-CSF alone [52]. Furthermore, the use of chemo-immunotherapy (i.e., Rituximab) to mobilize stem cells resulted in delayed lymphocyte recovery post-ASCT [53]. These reports suggest that chemo-mobilization could be detrimental to the mobilization and collection of immune effector cells.

Plerixafor: Plerixafor is a CXC4 antagonist approved in combination with G-CSF for the mobilization of stem cells. The CXCL12/CXCR4 is a chemokine signal trafficking T and NK cells. Plerixafor has been reported to mobilize dendritic cells, CD3, CD4, CD8, and NK cells [54].

### 10.2. Collection of Immune Effector Cells

Number of apheresis collections: The number of apheresis collections has been associated with higher lymphocyte collections. Collections from patients who received four apheresis collections or more resulted in higher autograft-absolute lymphocyte count numbers. The problems with this approach are cost and patient satisfaction [52].

Manipulation of apheresis machines: Our apheresis group has been able to change the settings of the Fenwal Amicus machine to augment the collection of lymphocytes. It is our standard practice at the Mayo Clinic, Rochester, MN that any lymphoma patient undergoing ASCT will not only have enough CD34 stem cells (4.0 × 10^6^ cells/kg) collected to facilitate hematologic engraftment but also A-ALC ≥ 0.5 × 10^9^ cells/kg to impact ALC-15 and survival post-ASCT [55].

Autograft immuno-suppressor depletion: Another strategy to develop an autologous graft-versus-tumor effect is to deplete the immunosuppressive cells collected with the stem cells in the autograft. An example of would the depletion of CD14+HLA-DR^DIM^ cells using the CD14+ beads column from the autograft [56].

## 11. Conclusions

In this review, I summarized our current understanding of the type of immune effector cells that are mobilized, collected, and infused from the autograft in patients undergoing ASCT. The published reports indicate that the infusion of autograft absolute numbers of immune effector cells directly impact the clinical outcomes post-ASCT. Thus, new strategies are warranted to mobilize, collect, and infuse autograft immune effector cells (i.e., dendritic cells type 1, CD4, CD4-PD-1-, CD8, natural killer cells, and NKp30 cells) that improve survival. Furthermore, methods to minimize the collection and infusion of immuno-suppressive cells (i.e., macrophage type 2, myeloid-derived suppressor cells, CD14+HLA-DR^DIM^) are also needed, as these immuno-suppressor cells negatively impact survival outcomes post-ASCT. The clinical outcome benefits observed based on the specific type of infused autograft immune effectors support the concept of an autologous graft versus tumor effect; thus, they indicate that ASCT should be seen not only as a means to give high-dose chemotherapy to treat cancer, but also as an immunotherapeutic treatment.

## Figures and Tables

**Table 1 cells-11-02197-t001:** Clinical outcomes post-autologous stem cell transplantation based on the infusion of autograft-absolute lymphocyte count.

Disease [Study Reference]	Autograft Immune Effector Cell	Median Time (months) *	5-Years OS Rates *	5-Years PFS Rates *	*p*-Value
MM [19]	A-ALC ≥ 0.5 × 10^9^ cells/kg	58.0	52%		<0.0002
	A-ALC < 0.5 × 10^9^ cells/kg	30.0	30%		
	A-ALC ≥ 0.5 × 10^9^ cells/kg	22.0		24%	<0.0001
	A-ALC < 0.5 × 10^9^ cells/kg	15.0		12%	
B-cell NHL [18]	A-ALC ≥ 0.5 × 10^9^ cells/kg	76.0	57%		<0.0001
	A-ALC < 0.5 × 10^9^ cells/kg	17.0	20%		
	A-ALC ≥ 0.5 × 10^9^ cells/kg	49.0		50%	<0.0001
	A-ALC < 0.5 × 10^9^ cells/kg	10.0		13%	

Abbreviations: A-ALC = autograft-absolute lymphocyte count; MM = multiple myeloma; NHL = Non-Hodgkin’s Lymphoma; OS = overall survival; PFS = progression-free survival. * The median times and the overall survival and the progression-free survival rates were calculated from the day of stem cell infusion (Day 0).

**Table 2 cells-11-02197-t002:** Clinical outcomes post-autologous stem cell transplantation based on the infusion of autograft-professional antigen-presenting cells.

Disease [Study Reference]	Autograft Immune Effector Cell	Median Time (Months)*	5 Years OS Rates *	5 Years PFS Rates *	*p*-Value
B-cell NHL [27]	DC ≥ 9.0 × 10^6^ cells/kg	Not reached	~60% (2 -years OS rates)		<0.022
	DC < 9.0 × 10^6^ cells/kg	11.5	~40% (2 -years OS rates)		
	DC1 ≥ 4.0 × 10^6^ cells/kg	Not reached	~55% (2 -years OS rates)		<0.04
	DC1 < 4.0 × 10^6^ cells/kg	11.3	~34% (2 -years OS rates)		
B-cell NHL [28]	DC1 ≥ 0.12 × 10^9^ cells/kg	Not reached		64%	<0.0001
	DC1 < 0.12 × 10^9^ cells/kg	23.9		30%	

Symbol: ~ = approximation of survival rates extrapolated from the Kaplan–Meier survival curves reported in the references. Abbreviation: DC = dendritic cells; DC1 = dendritic cells type 1; MM = multiple myeloma; NHL = Non-Hodgkin’s Lymphoma; OS = overall survival; PFS = progression-free survival. * The median times and the overall survival and the progression-free survival rates were calculated from the day of stem cell infusion (Day 0).

**Table 3 cells-11-02197-t003:** Clinical outcomes post-autologous stem cell transplantation based on the infusion of autograft-innate immune system cells.

Disease [Study Reference]	Autograft Immune Effector Cell	Median Time (Months) *	5 Years OS Rates *	5 Years PFS Rates *	*p*-Value
B-cell NHL [28]	Macrophages Type 2 ≥ 0.03 × 10^9^ cells	12.67		13%	<0.0001
	Macrophages Type 2 < 0.03 × 10^9^ cells	Not reached		77%	
MM [31]	A-NK ≥ 2.5 × 10^6^ cells/kg	Not reached	~92%		<0.006
	A-NK < 2.5 × 10^6^ cells/kg	Not reached	~70%		
B-cell NHL [32]	A-NK ≥ 0.09 × 10^9^ cells/kg	Not reached	87%		<0.0003
	A-NK < 0.09 × 10^9^ cells/kg	Not reached	55%		
	A-NK ≥ 0.09 × 10^9^ cells/kg	Not reached		71%	<0.0001
	A-NK < 0.09 × 10^9^ cells/kg	23.8		32%	
B-cell NHL [28]	A-KIR2DL2 (NK cells) ≥ 0.066 × 10^9^ cells/kg	48.2	33%		<0.0001
	A-KIR2DL2 (NK cells) < 0.066 × 10^9^ cells/kg	Not reached	88%		
B-cell NHL [28]	A-NKp30 (NK cells) ≥ 0.09 × 10^9^ cells/kg	Not reached	95%		<0.0001
	A-NKp30 (NK cells) < 0.09 × 10^9^ cells/kg	34.8%	34%		
	A-NKp30 (NK cells) ≥ 0.09 × 10^9^ cells/kg	Not reached		77%	<0.0001
	A-NKp30 (NK cells) < 0.09 × 10^9^ cells/kg	2.7		13%	

Abbreviations: A-NK = autograft-natural killer cells; A-KIR = autograft-killer cell immunoglobulin-like receptors; MM = multiple myeloma, NHL = Non-Hodgkin’s Lymphoma; OS = overall survival; PFS = progression-free survival. Symbol: ~ = approximation of survival rates extrapolated from the Kaplan–Meier survival curves reported in the references. * The median times and the overall survival and the progression-free survival rates were calculated from the day of stem cell infusion (Day 0).

**Table 4 cells-11-02197-t004:** Clinical outcomes post-autologous stem cell transplantation based on the infusion of autograft-adaptive immune system.

Disease [Study Reference]	Autograft Immune Effector Cell	Median Time (Months) *	5 Years OS Rates *	5 Years PFS Rates *	*p*-Value
MM [31]	A-CD3 ≥ 20 × 10^6^ cells/kg	Not reached			<0.017
	A-CD3 < 20 × 10^6^ cells/kg	47.0			
	A-CD8 > 15 × 10^6^ cells/kg	Not reached	~82%		<0.032
	A-CD8 ≤ 15 × 10^6^ cells/kg	Not reached	~65%		
MM [37]	A-CD4 ≥ 45% of gated apheresis lymphocytes	~3.8	~38%		<0.003
	A-CD4 < 45% of gated apheresis lymphocytes	~1.6	~10%		
B-cell NHL [36]	A-CD3 ≥ 23.1 × 10^6^ cells/kg	Not reached	~82%		<0.008
	A-CD3 < 23.1 × 10^6^ cells/kg	40	~40%		
B-cell NHL [38]	A-CD4 ≥ 37 × 10^6^ cells/kg	Not reached		~78%	<0.028
	A-CD4 < 37 × 10^6^ cells/kg	Not reached		~60%	
B-cell NHL [28]	A-CD4-PD-1- ≥ 0.13 × 10^9^ cells/kg	Not reached		67%	<0.0001
	A-CD4-PD-1- < 0.13 × 10^9^ cells/kg	23.0		30%	

Abbreviations: A-CD = autograft-cluster of differentiation; A-PD-1 = autograft-program cell death ligand 1; MM = multiple myeloma; NHL = Non-Hodgkin’s Lymphoma; OS = overall survival; PFS = progression-free survival. Symbol: ~ = approximation of survival rates extrapolated from the Kaplan–Meier survival curves reported in the references. * The median times and the overall survival and the progression-free survival rates were calculated from the day of stem cell infusion (Day 0).

**Table 5 cells-11-02197-t005:** Clinical outcomes post-autologous stem cell transplantation based on the infusion of autograft-immunosuppressive effector cells.

Disease [Study Reference]	Autograft Immune Effector Cell	Median Time (Months) *	5 Years OS Rates *	5 Years PFS Rates *	*p*-Value
B-cell NHL [41]	A-AMC ≥ 0.5 × 10^9^ cells/kg	25.5	40%		<0.0001
	A-AMC < 0.5 × 10^9^ cells/kg	157.5	65%		
	A-AMC ≥ 0.5 × 10^9^ cells/kg	12.8		34%	<0.0001
	A-AMC < 0.5 × 10^9^ cells/kg	151.0		62%	
B-cell NHL [41]	A-LMR ≥ 1	167.2	73%		<0.0001
	A-LMR < 1	17.6	30%		
	A-LMR ≥ 1	152.9		67%	<0.0001
	A-LMR< 1	6.6		26%	
T cell NHL [42]	A-LMR ≥ 1	Not reached	87%		<0.0001
	A-LMR < 1	17.9	26%		
	A-LMR ≥ 1	Not reached		72%	<0.0001
		11.9		16%	
B-cell NHL [28]	MDSC ≥ 0.31 × 10^9^ cells/kg	21.6	33%		<0.0001
	MDSC < 0.31 × 10^9^ cells/kg	Not reached	88%		
	MDSC ≥ 0.31 × 10^9^ cells/kg	7.1		19%	<0.0001
	MDSC ≥ 0.31 × 10^9^ cells/kg	Not reached		70%	
B-cell NHL [32]	A-CD14+HLA-DR^DIM^ ≥ 0.21 × 10^6^ cells/kg	73.1	52%		<0.0005
	A-CD14+HLA-DR^DIM^ < 0.21 × 10^6^ cells/kg	Not reached	83%		
	A-CD14+HLA-DR^DIM^ ≥ 0.21 × 10^6^ cells/kg	31.1		34%	<0.004
	A-CD14+HLA-DR^DIM^ < 0.21 × 10^6^ cells/kg	Not reached		62%	

Abbreviations: A-AMC = autograft-absolute monocyte count; A-LMR = autograft-lymphocyte to monocyte ratio; A-MDSC = autograft-myeloid-derived suppressor cell; HLA-DR = Human Leukocyte Antigen-DR; NHL = Non-Hodgkin’s Lymphoma; OS = overall survival; PFS = progression-free survival. * The median times and the overall survival and the progression-free survival rates were calculated from the day of stem cell infusion (Day 0).

## Data Availability

Not applicable.

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
