# Peer review of "The Impact of Infused Autograft Absolute Numbers of Immune Effector Cells on Survival Post-Autologous Stem Cell Transplantation"

_cells, 2022, doi:10.3390/cells11142197_

Round 1
Reviewer 1 Report
Page 3, line 125 - "...were 80% verus 50%", versus*
Author Response
Please see Response letter file

Reviewer 2 Report
This review describes the impact on the clinical outcomes based on the infusion of different kind of immune cells during autologous stem cell transplantation supporting the concept of autologous graft versus tumor effect. The role of immune effector cells in transplant is well documented. Although the review is comprehensive but does not provide any new approaches to the field.
Author Response
Please see response letter file

Reviewer 3 Report
Despite the complexity of the subject, this review regarding the immune effector cells that are collected and infused in the autograft and their impact with outcome measures is original, interesting and intriguing. It allows to discover a range of new potential modalities to improve mobilization strategies in order to increase the number of cells positively or negatively impacting survival in hematologic malignancies as Multiple Myeloma and B Cell non Hodgkin Lymphoma. It would be interesting to know to what extent new therapies used for multiple myeloma as immunomodulatory agents as lenalidomide or anti-CD38 monoclonal antibodies as daratumumab and isatuximab, recently included in the upfront regimens prior autologous stem cell transplantation, can affect the immune cells in the autograft. The paper is well written, scientifically accurate and the reference section is wide and appropriate.
Author Response
See Response Letter file

Reviewer 4 Report
The authors present interesting associations of graft composition and graft recovery with outcomes.
They suggest this can be explained but graft-versus-tumour effects of a autologous transplant. This is very speculative. There are other explanations for that e.g. lymphocyte recovery also marks graft function, disease status, patient performance and other factors.
The G-v-T would have to depend on cancer neoantigenes. Can the author explain how should these be recognized as foreign by T cell or other cells educated in their presence? It is not discussed at all.
Why should there by G-v-T if there is no G-v-H?
Why should myeloablation boost patients own immune response against cancer?
Is there any literature that show antigen-specific responses after autoSCT?
Author Response
See Response letter file

Reviewer 5 Report
Porrata L.F. presented an interesting review on the impact of infused autograft absolute numbers of immune effector cells on survival post-autologous stem cell transplantation.
Major Comment
The author adequately presented a very interesting issue, which can change the paradigm of viewing the role of Auto-HSCT.
Although the report is mainly characterized by corrected self-citation, it is clear that Auto-HSCT can have a future of an immunotherapeutic effect.
Just one question: it seems that effector cells can act after Auto-HSCT in a dose-dependent manner. Thus, in those patients who did not receive Auto-HSCT did you find a higher rate of effector cells, correlated to a better outcome?
row 254. Table 5, I suppose, instead of Table 1.
row 357. Please specify. It is unclear reading the sentence "in the allogeneic stem cell transplantation".
I would suggest to use abbreviation, as much as possible, in the text as well as in the Tables.
Minor comment
row 40 Pleas insert "....seen in the allogeneic...." (as well as in row 53)
row 78 Please erase the initial "With"
row 100 Please insert "....collected and infused..."
row 103 Please erase "in".....autograft....
row 125 Please correct "versus" instead of "verus".
row 141-142 Please insert "...seen in the post-autologous...."
row 147 please erase "cells" after type 1 at the beginning of the sentence, and erase the repetitive "dendritic cells type 1".
row 151 I would suggest to write as follow ".....dendritic cells type 1 to induce the development of cytotoxic T-cells..."
row 162 Please specify the abbreviation A-DC.
row 219 Program cell death ligand 1 should be abbreviate as PD-1 L.
row 229 "our group is the first...that published that....."
row 315 Ruxolitinib
row 316 Etanercept
row 330 I would use "targeted" instead of "desired".
Tables. Please reduce the character's dimension of legenda. Please specify the Median Times: from Day-0??
Table 2. Rows 4 and 5. Please check: DC Type 1 > are in both row. The second one (row 5), I suppose that It should be < than 4.0.
Author Response
See response letter file

Round 2
Reviewer 4 Report
The manuscript is now improved.